# Cross-sectional validation of the COPD Assessment Test (CAT) among chronic obstructive pulmonary disease patients in rural Uganda

Charles Batte[1,2], Andrew Weil Semulimi[1,3]*, Ronald Kasoma Mutebi[1,4,5], Nelson Twinamasiko[1,4], Sarah Racheal Muyama[1], John Mukisa[6], Immaculate Atukunda[7], David Mukunya[8], Robert Kalyesubula[3,9], Siddharthan Trishul[10], Bruce Kirenga[1,11]

**1** Department of Medicine, Lung Institute, School of Medicine, College of Health Sciences, Makerere University, Kampala, Uganda, **2** Climate Change and Health Unit, Tree Adoption Uganda, Kampala, Uganda, **3** Department of Physiology, School of Biomedical Sciences, College of Health Sciences, Makerere University, Kampala, Uganda, **4** Clinical Epidemiology Unit, School of Medicine, College of Health Sciences, Makerere University, Kampala, Ugand, **5** Department of Medicine, Mengo Hospital, Kampala, Uganda, **6** Department of Immunology and Molecular Biology, School of Biomedical Sciences, College of Health Sciences, Kampala, Uganda, **7** Department of Ophthalmology, School of Medicine, College of Health Sciences, Makerere University, Kampala, Uganda, **8** Faculty of Health Sciences, Department of Community and Public Health, Busitema University, Mbale, Uganda, **9** African Community Centre for Social Sustainability (ACCESS), Nakaseke, Uganda, **10** Division of Pulmonary, Allergy, Critical Care, and Sleep, Miller School of Medicine, University of Miami, Miami, Florida, United States of America, **11** Department of Medicine, School of Medicine, College of Health Sciences, Makerere University, Kampala, Uganda

* andrewweil89@gmail.com

**Data Availability Statement:** The datasets used and/or analysed are available in Figshare via the link: https://doi.org/10.6084/m9.figshare.22578463.

## Abstract

Measuring quality of life is a key component in the management of Chronic Obstructive Pulmonary Disease (COPD). The COPD assessment test (CAT), an easy to administer and shorter instrument than the standard Saint George's respiratory questionnaire (SGRQ), could be an alternative tool for measuring the quality of life of COPD patients in rural Uganda. A cross-sectional study was conducted between June and August 2022, consecutively recruiting 113 COPD patients aged > 40 years from the Low-Dose Theophylline for the management of Biomass-associated COPD (LODOT-BCOPD) study. Upon obtaining consent, participants answered an interviewer administered social demographic, CAT and SGRQ questionnaire. Internal consistency for both SGRQ and CAT was determined using Cronbach's alpha coefficient and values > 0.7 were considered acceptable while correlations were determined using Spearman's rank correlation. Limits of Agreement were visualised using Bland Altman and pair plots. Of the 113 participants, 51 (45.1%) were female. The mean age was 64 ± 12 years, 19 (16.8%) had history of smoking while majority (112 (99.1%)) reported use of firewood for cooking. There was a strong correlation of 0.791 (p < 0.001) between the CAT and SGRQ total scores with a high internal consistency of CAT, Cronbach's alpha coefficient of 0.924 (0.901–0.946). The agreement between the absolute CAT scores and the SGRQ scores was good with a mean difference of -0.932 (95% Confidence Interval: -33.49–31.62). In summary, CAT has an acceptable validity and can be

**Funding:** The study was funded through the American Thoracic Society 2021 MECOR Research Award (awarded to CB) and the Makerere University Non-Communicable Diseases (MAKNCD) Research Training Program: supported by the Fogarty International Center of the National Institutes of Health under Award Number D43TW011401 (awarded to BK and WC). The content is solely the responsibility of the authors and does not necessarily represent the official views of the National Institute of Health or the American Thoracic Society. CB, AWS, RKM, NT, SRM, JM and DM received salary from the MECOR 2021 Research award. CB and BK receive salary from the Makerere University Non-Communicable Diseases (MAKNCD) Research Training Program. The funders had no role in study design, data collection and analysis, decision to publish, or preparation of the manuscript.

**Competing interests:** The authors have declared that no competing interests exist.

used as an alternative to the SGRQ to assess the quality of life of COPD patients in rural Uganda.

## Introduction

In 2019, 6% (3.23 million) of global mortality was attributable to chronic obstructive pulmonary disease (COPD) [1,2] and by 2030, an increase of 7.8% in COPD related global mortality is expected [3]. Coincidentally, in 2030, over 250,074 individuals are likely to be diagnosed with COPD which is a 155% increase from 98,368 in 2010 [4]. Sub- Saharan Africa with a high prevalence of infectious diseases such as HIV and tuberculosis that are important risk factors of COPD [5,6], has a prevalence of COPD ranging from 4.1 to 24.8% [7–9] which translates to 26.3 million (18.5 to 43.4 million) [9] indicating a significant dual burden of disease. In Uganda, the prevalence of COPD is estimated at 1.5% to 16.2% [10,11] which is likely to increase in the near future [10] due to the increasing trends of urbanisation and other risk factors such as continued use of biomass fuels in Uganda [10,12].

The rising prevalence of COPD in Uganda is worrisome due to the already strained health system. The chronic nature and acute flare-ups of COPD necessitate meticulous follow-up and continuous presence of health workers who might not be readily available in rural areas of Uganda. Effective management strategies and tools for monitoring impact of therapeutic interventions are essential to facilitate the restoration to normal daily activities, prevent acute flare-ups and enhance productivity among COPD patients and promote psychological-behavioural changes such as cessation of smoking [13]. Assessment of the quality of life of COPD patients has become a mainstay in the management of COPD patients [13] since quality of life has been shown to significantly predict impaired Force Expiratory Volume in one second ($FEV_1$) and disease severity [14,15]. Different COPD-specific quality of life measurements for example the St. George's Respiratory Questionnaire (SGRQ) have been validated and used widely in clinical settings to determine the quality of life of COPD patients [16–19]. The SGRQ tool has been adopted and widely used by different countries in various clinical settings including Uganda [19]. However, its complexity, time-consumption and requirement of special training hinder its routine use and conduct especially in Low–and middle–income countries (LMICs) like Uganda [20].

COPD Assessment Tool (CAT), was developed to improve communication between the healthcare provider and the patient and provide a reliable assessment of the health status of the patient [21]. CAT has been translated and validated for use in clinical setting [20,22,23]. Moreover, studies have showed that CAT scores positively correlate with SGRQ scores [24] which makes it a good and valid tool to measure quality of life of COPD patients. In Uganda, quality of life among COPD patients has mostly been assessed using SGRQ [17,19]. In this study, we explored the validity of the CAT and assessed its internal consistency and limits of agreement as compared to the SGRQ.

## Methods

### Study design

This was a cross-sectional study conducted between June and August 2022.

### Study setting

The study was conducted at the out-patient clinic of Nakaseke General Hospital found in Nakaseke, which is 50 km from Kampala. The clinic is manned by approximately seven health

workers who review about 20 patients with COPD per week and the ratio of doctors and nurses per person in Nakaseke health sub-district was 1:25,000 and 1:5000, respectively [25]. The hospital serves over 43,167 households with an estimated population of 208,500 [10,26]. Luganda the local language used to conduct the interviews is the predominant language in the health sub–district. Over 75% of the people living within Nakaseke are subsistence farmers and majority of them cook with firewood.

## Study population

Participants who were more than 40 years old and diagnosed with COPD by spirometry as per the American Thoracic Society/ European Respiratory Society guidelines [27] were recruited from the Low-Dose Theophylline for the management of Biomass-associated COPD (LODOT-BCOPD) trial which is described elsewhere [28] (**Fig 1**).

## Sample size estimation

We recruit 113 participants. The sample size was estimated based on the formulas and equations by Bonnet [29] for sample sizes involving determining Cronbach's α where we assumed that k = 8 (number of items in the CAT), alpha coefficient of 0.732 [30] at a 95% confidence

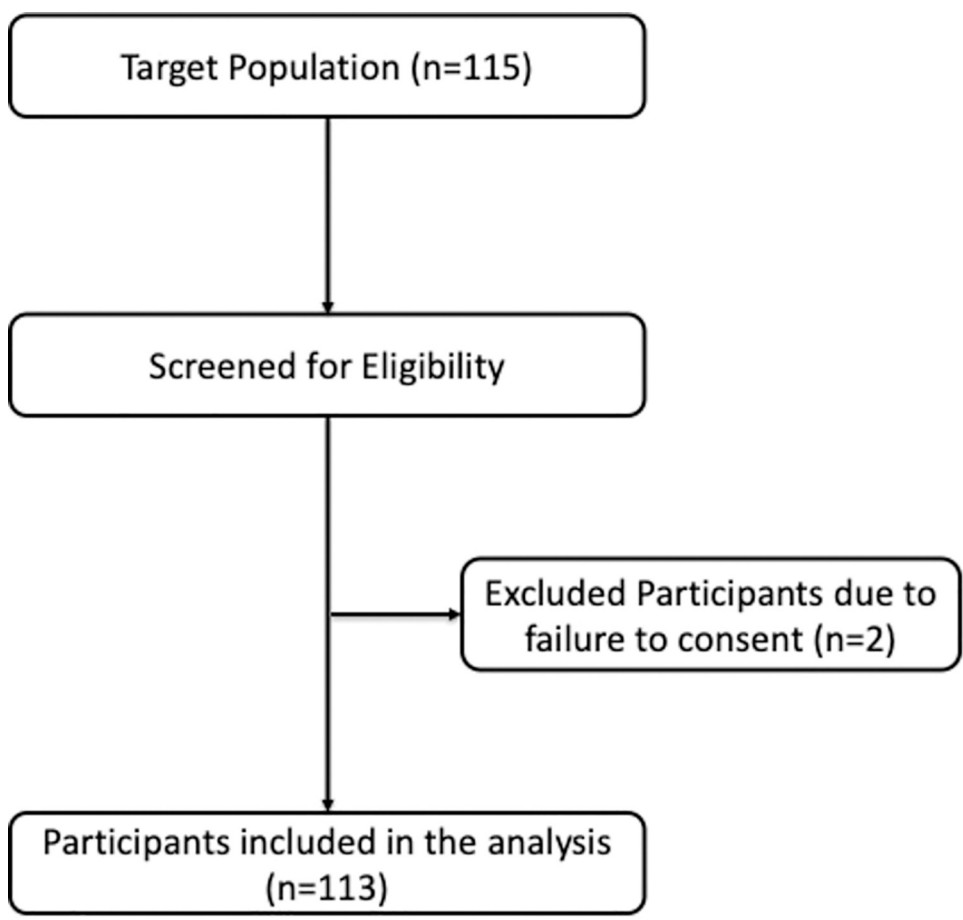

**Fig 1. The study flow chart highlighting how participants were recruited.**

interval and a assumed validated effect size parameter, δ of 1.5. We used consecutive sampling to recruit 113 participants from the LODOT-BCOPD trial.

## Study procedures

Prospective participants were invited for a study interview by the study nurse or medical doctor. Upon completion of the informed consent process, participants were interviewed in their language of preference by the study nurse or doctor but all of them chose Luganda. During the interview, participants answered questions from an investigator designed socio-demographic questionnaire, CAT and SGRQ.

## Data collection tool

The socio-demographics questionnaire assessed for age, sex, number of individuals in a household, risk factors such as biomass exposure history (firewood), smoking history, and presence of comorbidities such as hypertension, previous history of Tuberculosis, HIV, and asthma. The main outcome variables were limits of agreement and Cronbach's Alpha.

CAT is an eight items tool which assess for cough, sputum production, chest tightness, dyspnea lack of energy and sleep disturbance [22]. In addition, it assesses for limitation in doing activities at home and confidence leaving home. Its scale ranges from 1 to 5 with a total score of 0 to 40. The GOLD 2021 report considers a threshold CAT score above 10 to be symptomatic and requires treatment.

SGRQ is a 50-item disease-specific instrument designed to measure impact on overall health, daily life, and perceived well-being in patients with obstructive airways disease such as COPD [31]. Its items are weighted and categorized in 2 parts such as one with symptoms component and the other with activities that cause or are limited by breathlessness. Completion of this tool takes approximately 8 to 15 minutes. The scores range from 0 to 100 with the highest scores indicative of limitations. This was considered the gold standard in this study.

The study used a Luganda version of the SGRQ which was validated in a previous study conducted in Uganda [32]. The official Luganda version of CAT questionnaire was provided by MAPI research trust, PROVIDE, 27 rue de la Villette, 69003 Lyon, France who are authorised to provide translated versions of the CAT questionnaire upon request [33]. Prior permission was sought to use both questionnaires.

## Statistical analysis

The data were analysed using Stata, version 16. 0 (StataCorp LLC, College Station, TX, USA). The categorical variables were summarized as frequencies and percentages while continuous variables as median and interquartile range or mean and standard deviation as appropriate. The SGRQ total scores were calculated using the SGRQ calculator. Internal consistency for both SGRQ and CAT was determined using Cronbach's alpha coefficient with corresponding confidence intervals [34,35] and values > 0.7 were considered acceptable. Data was assessed for normality using the Shapiro–Wilk test and correlation analysis was done using Spearman's rank correlation coefficient. We compared absolute SGRQ and CAT scores using Bland Altman and pair plots to visualize the limits of agreement between the two tools. We transformed the CAT scores whose maximum score is 40 by a multiple 2.5 to match the maximum SGRQ scores of 100 hence making them comparable [36]. It is recommended that 95% of the data points should lie within 2 standards deviations of the mean differences between the two measurements.

## Ethical consideration

Ethical approval was sought from the Infectious Disease Institute Research and Ethics Committee (IDIREC Ref: 045/2022) and the Uganda National Council for Science and Technology (HS2145ES) while administrative clearance was sought from Nakaseke Hospital and District leadership. We obtained written informed consent from participants before they were recruited into the study. Participants who did not provide written informed consent were excluded from the study. Confidentiality and privacy of the participants was strictly observed in this study and all study procedures were conducted in in accordance with the Ugandan laws and regulation, Good Clinical Practice, and the principles of the Declaration of Helsinki.

## Results

### Participant characteristics

A total of 113 participants were recruited from the LODOT-BCOPD (**Fig 1**). The mean age was 64 ± 12 years, 62 (54.9%) of the participants were male and 19 (16.8%) reported history of smoking cigarettes averaging 6 ± 5 cigarettes per day. The median number of individuals per household was 4 (2–6), and almost all the participants 112 (99.1%) used firewood as a major source of cooking fuel. Hypertension was the most reported co-morbidity at 43 (38.1%). (**Table 1**).

### Validity

The responses to the CAT and SGRQ were obtained from the same participants. The CAT and SGRQ had high Cronbach's alpha coefficient of 0.924 (0.901–0.946) and 0.947 (0.937–0.958) respectively which shows significant internal consistency (**Table 2**). The Spearman's correlation was recorded between the CAT and SGRQ total scores at 0.791, p value < 0.001 (**Fig 2**).

The correlation between CAT score and domains of SGRQ was significant with 0.701, 0.713 and 0.782, p value < 0.001, for symptom score, activity score and impact score respectively. The agreement between the absolute CAT scores and the SGRQ scores was good with a mean difference of -0.932 (95% Confidence Interval: -33.49–31.62) (**Figs 3 and 4**).

## Discussion

In this study, the CAT and SGRQ total scores had a strong correlation, 0.791 and CAT had a high internal consistency. The agreement between the two tools as per the Bland Altman score was significant demonstrating no systematic bias.

The findings from this study are comparable to results from other studies; for instance the first version of CAT had a Cronbach's alpha coefficient of 0.88 [37], Tsuda and colleagues reported Cronbach's alpha coefficient = 0.891 using the Japanese version [38] while Pothirat and colleagues reported Cronbach's alpha coefficient = 0.853 using the Thai version [39]. In another study conducted among 90 Greek COPD patients, the Cronbach's alpha coefficient of CAT was 0.86 [40] and a systematic review which included studies from several countries in Europe, North America, South America, Asia and Africa, the internal consistency of CAT reported (Cronbach's alpha coefficient = 0.85–0.98) [41]. The results from this study were consistent with findings from previous studies which showed moderate to strong correlation between CAT and SGRQ total scores and domains [37,39,41–43]. The high Cronbach's alpha score and strong correlation recorded in this study could be attributed to the differences in the sampled population in terms of presence of co-morbidities and age.

In terms of limits of agreement, the Bland Altman and pair plot revealed a significant relationship between the two tools, which is consistent with findings from the *Paul et al* [44] and

**Table 1. The characteristics of participant recruited into the study.**

| Variable | n (%) |
|---|---|
| **Sex** | |
| Female | 51 (45.1) |
| Male | 62 (54.9) |
| **Age (years)** *mean ± SD* | 64 ±12 |
| **Marital status** | 65 (12) |
| Single | 16 (14.2) |
| Divorced | 17 (15.0) |
| Widowed | 26 (23.0) |
| Married/cohabiting | 54 (47.8) |
| **Education status** | |
| None | 47 (41.6) |
| Primary | 56(49.6) |
| Secondary | 10 (8.8) |
| **Employment status** | |
| No employment | 43 (38.0) |
| Informal employment | 32 (28.3) |
| Self-employment | 36 (31.9) |
| Formal employment | 2 ((1.8) |
| **Risk factors of COPD** | |
| **Number of individuals in a household** *median (IQR)* | 4 (2–6) |
| **History of smoking,** *yes* | 19 (16.8) |
| **Smoked daily** n = 19 | 17 (89.5) |
| **Number of cigarettes per day** n = 19 *mean ± SD* | 6 ±5 |
| **Use of fumigants** *Yes* | 23 (20.4) |
| **Frequency of fumigant** n = 23 *median (IQR)* | 2 (1–3) |
| **Fuel used in cooking** | |
| Firewood, *yes* | 112 (99.1) |
| Gas, *yes* | 1 (0.9) |
| Electricity, *yes* | 1 (0.9) |
| Charcoal, *yes* | 53 (46.9) |
| **Activities other than cooking** | |
| Local brewing, *yes* | 4 (3.5) |
| Charcoal burning, *yes* | 14 (12.4) |
| Brick laying, *yes* | 6 (5.3) |
| **Co-morbidities** | |
| Tuberculosis, *yes* | 12 (10.6) |
| HIV, *yes* | 20 (17.7) |
| Hypertension, *yes* | 43 (38.1) |
| **Previous chronic illnesses** | |
| Tuberculosis, *yes* | 33 (29.2) |

n—number of participants, SD—standard deviation, IQR—interquartile range.

*Tsiligianni at al* [40] studies. Although SGRQ is considered the gold standard for assessing patient symptoms by the GOLD guidelines [13], it is a technical and long question which limits its application in inadequately staffed clinical settings like those in LMICs. Results from this

**Table 2. Validity of CAT scores.**

| Cronbach's Alpha Coefficient and score distribution of the CAT and SGRQ tools | | | | | |
|---|---|---|---|---|---|
| Tool | Score range | Items (n) | Cronbach's coefficient (95% CI) | Mean ± SD | Median (IQR) |
| CAT | 0–40 | 8 | 0.924 (0.901–0.946) | 19.11 ± 10.84 | 20 (0–39) |
| SGRQ | 0–100 | 50 | 0.947 (0.937–0.958) | 48.7 ± 22.24 | 50.41 (3.47–88.01) |
| Correlation between CAT and SGRQ scores | | | | | |
| | SGRQ Total score | SGRQ Symptom score | SGRQ Activity score | SGRQ Impact score | |
| CAT Score | 0.791 | 0.701 | 0.713 | 0.782 | P value < 0.001 |

study show that CAT could be a shorter and easy to use alternative questionnaire that can be used to assess the quality of life of patients with COPD in rural Uganda.

However, this study has limitations that may delay its adoption in Ugandan clinical settings. The study was cross-sectional in nature which made calculations for repeatability through the intraclass correlation coefficient impossible. In the LODOT-BCOPD trial, theophylline, the trial drug is given to participants at different doses which may affect their quality of life subsequently the results of this study. The relationship of CAT with other parameters used to assess for COPD severity such as $FEV_1$, forced vital capacity (FVC) were not explored which provides an opportunity for future research. In addition, it highlights that CAT may not be used singly to assess the health status of COPD patients but in combination with other clinical parameters. Despite these limitations, this is the first study to validate CAT in rural Uganda and provides a basis for the utilisation of the CAT as a reliable alternative to SGRQ.

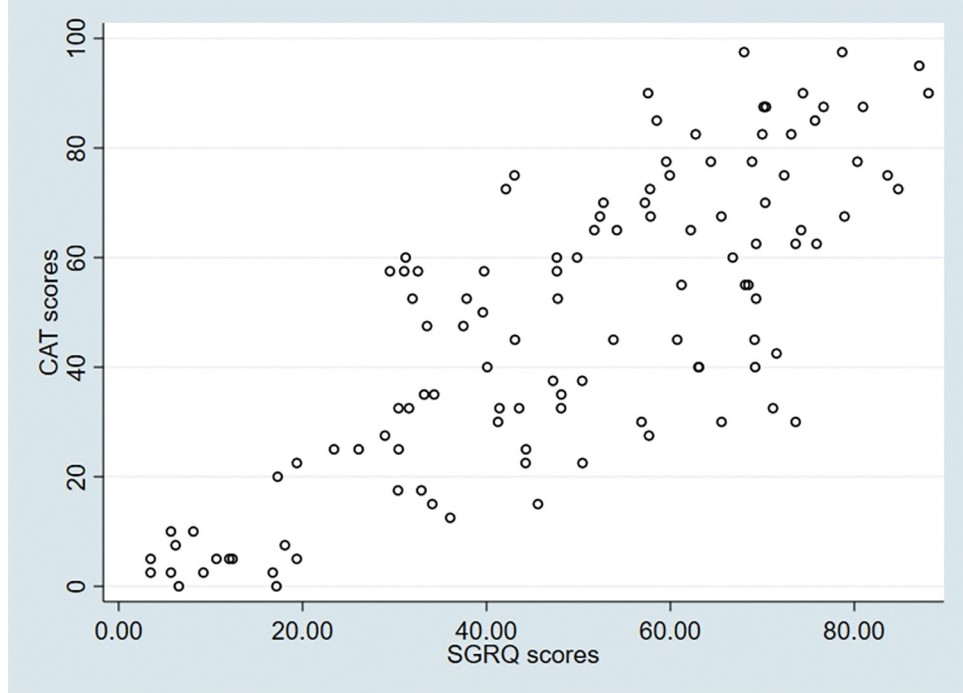

**Fig 2. Correlation between scores in St George's Respiratory Questionnaire (SGRQ) and COPD Assessment Test (CAT) in 113 participants.**

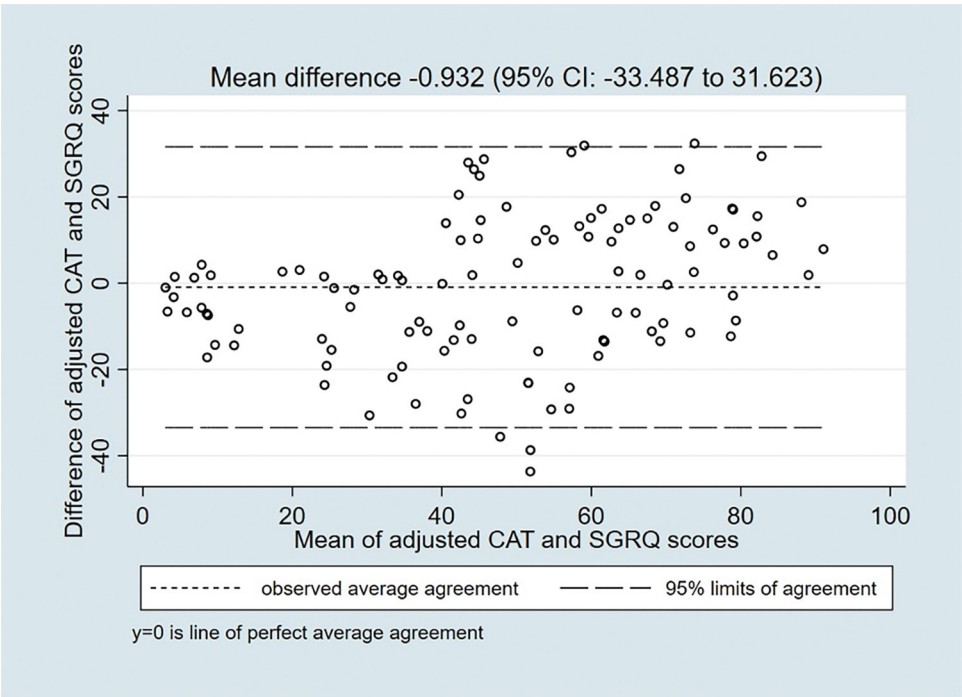

**Fig 3. Comparison between the difference of CAT scores and SGRQ scores and the means of CAT scores and SGRQ scores.**

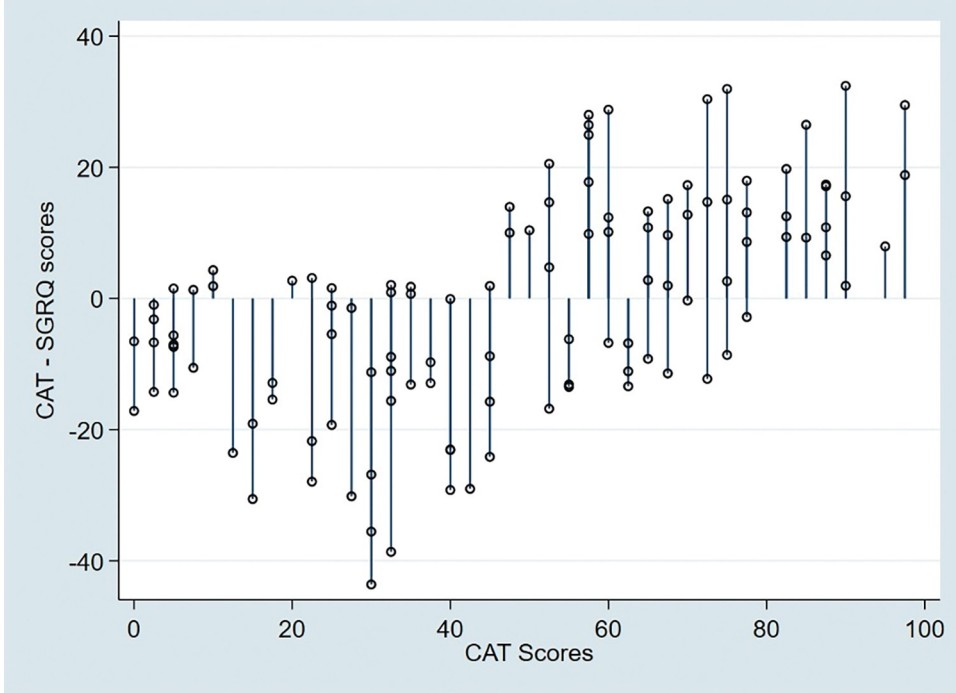

**Fig 4. A pair plot between the CAT- SGRQ scores and CAT scores.**

## Conclusion

CAT is a valid tool with high reliability that could be used as an alternative to SGRQ to measure the quality of life of COPD patients. We recommended that in rural and busy clinical settings, clinicians could use CAT in synergy with other clinical assessment to assess for health status of COPD patients.

## Author Contributions

**Conceptualization:** Charles Batte, Andrew Weil Semulimi, Ronald Kasoma Mutebi, John Mukisa, David Mukunya.

**Data curation:** Charles Batte, Andrew Weil Semulimi, Sarah Racheal Muyama, John Mukisa.

**Formal analysis:** Charles Batte, Ronald Kasoma Mutebi, John Mukisa, David Mukunya.

**Funding acquisition:** Charles Batte, Andrew Weil Semulimi, David Mukunya, Bruce Kirenga.

**Investigation:** Andrew Weil Semulimi, Nelson Twinamasiko, Sarah Racheal Muyama, John Mukisa, David Mukunya, Siddharthan Trishul.

**Methodology:** Charles Batte, Andrew Weil Semulimi, Ronald Kasoma Mutebi, Nelson Twinamasiko, John Mukisa, Immaculate Atukunda, David Mukunya, Robert Kalyesubula, Siddharthan Trishul, Bruce Kirenga.

**Project administration:** Charles Batte, Andrew Weil Semulimi.

**Resources:** Ronald Kasoma Mutebi, Nelson Twinamasiko, Sarah Racheal Muyama.

**Software:** Ronald Kasoma Mutebi.

**Supervision:** Andrew Weil Semulimi, Nelson Twinamasiko, Sarah Racheal Muyama, David Mukunya, Bruce Kirenga.

**Validation:** Charles Batte, Andrew Weil Semulimi, Nelson Twinamasiko, Sarah Racheal Muyama, John Mukisa, Immaculate Atukunda, David Mukunya, Robert Kalyesubula, Siddharthan Trishul.

**Visualization:** Charles Batte, Andrew Weil Semulimi, Ronald Kasoma Mutebi, Nelson Twinamasiko, John Mukisa, Immaculate Atukunda, David Mukunya, Robert Kalyesubula, Bruce Kirenga.

**Writing – original draft:** Andrew Weil Semulimi, Ronald Kasoma Mutebi, Siddharthan Trishul.

**Writing – review & editing:** Charles Batte, Andrew Weil Semulimi, Ronald Kasoma Mutebi, Nelson Twinamasiko, Sarah Racheal Muyama, John Mukisa, Immaculate Atukunda, David Mukunya, Robert Kalyesubula, Siddharthan Trishul, Bruce Kirenga.

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
