## [Decision Letter · Decision Letter 0]

22 Mar 2023

PGPH-D-23-00157

Cross-sectional validation of the COPD Assessment Test (CAT) among chronic obstructive pulmonary disease patients in rural Uganda.

Dear Dr. Semulimi,

Thank you for submitting your manuscript to PLOS Global Public Health. After careful consideration, we feel that it has merit but does not fully meet PLOS Global Public Health’s publication criteria as it currently stands. Therefore, we invite you to submit a revised version of the manuscript that addresses the points raised during the review process.

Please address all the reviewers' comments, particularly the issue with the language(s) used to administer the questionnaires.

We look forward to receiving your revised manuscript.

Kind regards,

Andre F. S. Amaral, Ph.D.

Academic Editor

Journal Requirements:

1. Please provide separate figure files in .tif or .eps format only and remove any figures embedded in your manuscript file. Please also ensure that all files are under our size limit of 10MB.

2. We do not publish any copyright or trademark symbols that usually accompany proprietary names, eg  ©, ®, ™  (e.g. next to drug or reagent names). Please remove all instances of trademark/copyright symbols throughout the text, including ™ on page 17.

Reviewers' comments:

Reviewer's Responses to Questions

**Comments to the Author**

1. Does this manuscript meet PLOS Global Public Health’s publication criteria? Is the manuscript technically sound, and do the data support the conclusions? The manuscript must describe methodologically and ethically rigorous research with conclusions that are appropriately drawn based on the data presented.

Reviewer #1: Yes

Reviewer #2: Partly

2. Has the statistical analysis been performed appropriately and rigorously?

Reviewer #1: Yes

Reviewer #2: I don't know

3. Have the authors made all data underlying the findings in their manuscript fully available (please refer to the Data Availability Statement at the start of the manuscript PDF file)?

Reviewer #1: Yes

Reviewer #2: No

4. Is the manuscript presented in an intelligible fashion and written in standard English?

Reviewer #1: Yes

Reviewer #2: Yes

5. Review Comments to the Author

Reviewer #1: This study is interesting and aims to test an alternative tool for assessing the health status of COPD affecting the quality of life. Some previous published articles conducted in other countries, e.g. the UK, Japan, Thailand and Greece, are similar to this study. However, I have some questions for the authors to improve the work as follows.

Introduction: The authors mentioned the high prevalence of HIV in Sub-Saharan Africa without showing any link with COPD which is the main outcome of interest.

Methods (study setting): The authors only presented the setting in the area of ‘Nakasake’, BUT not the clinic or the LODOT-BCOPD trial which may be relevant to this study.

Methods (study procedures): Participants were interviewed conveniently in either English or a local dialect – Luganda. Use of different languages might affects the authors’ ability to accurately convey the meaning of the data

Results and discussion: The authors recruited participants who were diagnosed with COPD; however, there are no details regarding the COPD staging (e.g. GOLD) which, itself, is related to the quality of life among COPD patients. Moreover, Table 1 presents participant characteristics; however, the authors did not present or discuss how these data affected the findings of the CAT/SGRQ test outcomes.

Reviewer #2: The study replicates a study of Validation of CAT in Uganda. This has been done all over the world and not necessarily novel, though it could be argued to be novel in Uganda. It is a study that validates the use of CAT scoring system for measuring quality of life, and has the advantage of its simplicity for use in in clinic settings.

As per the translation, it seem that the CAT and SGRQ questionnaires were not translated to the local language via standards processes of translating a questionnaire via bi-directional processes for consistency in language especially in a self administered tool. This creates a bias in interpretation depending on the ability of the transcriber or interpreter who reads out the questions. This could potentially be an issue as per the reproducibility of this study and the findings may vary depending on the ability of the interpreter.

6. PLOS authors have the option to publish the peer review history of their article (what does this mean?). If published, this will include your full peer review and any attached files.

**Do you want your identity to be public for this peer review?** For information about this choice, including consent withdrawal, please see our Privacy Policy.

Reviewer #1: No

Reviewer #2: No

---

## [Decision Letter · Decision Letter 1]

2 May 2023

PGPH-D-23-00157R1

Cross-sectional validation of the COPD Assessment Test (CAT) among chronic obstructive pulmonary disease patients in rural Uganda.

Dear Dr. Semulimi,

Thank you for submitting your manuscript to PLOS Global Public Health. After careful consideration, we feel that it has merit but does not fully meet PLOS Global Public Health’s publication criteria as it currently stands. Therefore, we invite you to submit a revised version of the manuscript that addresses the points raised during the review process.

The aim of the study was to compare and validate the CAT questionnaire against the SGRQ questionnaire. However, in lines 140 to 144, it is said that "The validated Luganda version of the CAT questionnaire..." was used – please clarify this as it sounds as if the CAT in Luganda had been validated prior to this study.

Citations of previous publications should be made only when relevant. Please remove:

- reference 12 from line 70 as this paper is on asthma and allergic rhinitis, not on COPD.

- references 12, 14 and 15 from line 72 as these papers focus on asthma/allergic rhinitis, tuberculosis and malnutrition and do not provide data/findings supporting your statement on biomass or urbanisation.

Please revise the English, as there are several sentences where words seem to be either missing or verbs incorrectly conjugated (for example, in line 140 “The study used a validated Luganda version of the SGRQ that had undergo the….”)

Please make sure that the files for submission include only the most recent clean and tracked changes versions.

We look forward to receiving your revised manuscript.

Kind regards,

Andre F. S. Amaral, Ph.D.

Academic Editor

Journal Requirements:

b. If any authors received a salary from any of your funders, please state which authors and which funders.

3. We do not publish any copyright or trademark symbols that usually accompany proprietary names, eg  ©, ®, ™  (e.g. next to drug or reagent names). Please remove all instances of trademark/copyright symbols throughout the text, including ™ on pages 8 and 19.

Reviewers' comments:

Reviewer's Responses to Questions

**Comments to the Author**

1. If the authors have adequately addressed your comments raised in a previous round of review and you feel that this manuscript is now acceptable for publication, you may indicate that here to bypass the “Comments to the Author” section, enter your conflict of interest statement in the “Confidential to Editor” section, and submit your "Accept" recommendation.

Reviewer #1: All comments have been addressed

2. Does this manuscript meet PLOS Global Public Health’s publication criteria? Is the manuscript technically sound, and do the data support the conclusions? The manuscript must describe methodologically and ethically rigorous research with conclusions that are appropriately drawn based on the data presented.

Reviewer #1: Yes

3. Has the statistical analysis been performed appropriately and rigorously?

Reviewer #1: Yes

4. Have the authors made all data underlying the findings in their manuscript fully available (please refer to the Data Availability Statement at the start of the manuscript PDF file)?

Reviewer #1: Yes

5. Is the manuscript presented in an intelligible fashion and written in standard English?

Reviewer #1: Yes

6. Review Comments to the Author

Reviewer #1: (No Response)

7. PLOS authors have the option to publish the peer review history of their article (what does this mean?). If published, this will include your full peer review and any attached files.

**Do you want your identity to be public for this peer review?** For information about this choice, including consent withdrawal, please see our Privacy Policy.

Reviewer #1: No

---

## [Editor Report · Decision Letter 2]

11 May 2023

Cross-sectional validation of the COPD Assessment Test (CAT) among chronic obstructive pulmonary disease patients in rural Uganda.

PGPH-D-23-00157R2

Dear Dr Semulimi,

We are pleased to inform you that your manuscript 'Cross-sectional validation of the COPD Assessment Test (CAT) among chronic obstructive pulmonary disease patients in rural Uganda.' has been provisionally accepted for publication in PLOS Global Public Health.

Best regards,

Andre F. S. Amaral, Ph.D.

Academic Editor